# Newborn Screening for Metachromatic Leukodystrophy in Tuscany: The Paradigm of a Successful Preventive Medicine Program

**DOI:** 10.3390/ijns11020030

**Published:** 2025-04-24

**Authors:** Sabrina Malvagia, Alessandra Bettiol, Margherita Porcaro, Massimo Mura, Silvia Funghini, Daniela Ombrone, Giulia Forni, Emanuela Scolamiero, Filippo Coppi, Roberta Damiano, Cristina Cereda, Simonetta Simonetti, Annalisa Lonetti, Marta Daniotti, Anna Caciotti, Amelia Morrone, Valeria Calbi, Francesca Fumagalli, Alessandro Aiuti, Elena Procopio, Renzo Guerrini, Giancarlo la Marca

**Affiliations:** 1Newborn Screening, Clinical Biochemistry and Clinical Pharmacy Laboratory, Meyer Children’s Hospital IRCCS, 50134 Florence, Italy; sabrina.malvagia@meyer.it (S.M.);; 2Department of Experimental and Clinical Biomedical Sciences “Mario Serio”, University of Florence, 50134 Florence, Italy; 3Center of Functional Genomics and Rare Diseases, Department of Pediatrics, Buzzi Children’s Hospital, 20154 Milan, Italy; 4Clinical Pathology and Neonatal Screening, Pediatric Hospital Giovanni XXIII, Azienda Ospedaliera Policlinico Giovanni XXIII, 70126 Bari, Italy; 5IRCCS Azienda Ospedaliero-Universitaria di Bologna, 40138 Bologna, Italy; 6Metabolic and Muscular Unit, Meyer Children’s Hospital IRCCS, 50134 Florence, Italy; 7Laboratory of Molecular Genetics of Neurometabolic Diseases, Neuroscience and Human Genetics Department, Meyer Children’s Hospital IRCCS, 50134 Florence, Italy; 8Department of Neuroscience, Psychology, Drug Research and Child Health (NEUROFARBA), University of Florence, 50134 Florence, Italy; 9San Raffaele Telethon Institute for Gene Therapy (SR-Tiget), IRCCS San Raffaele Scientific Institute, 20132 Milan, Italy; 10Pediatric Immunohematology Unit and BMT Program, IRCCS San Raffaele Scientific Institute, 20132 Milan, Italy; 11Neurology Unit and Neurophysiology Service, IRCCS San Raffaele Scientific Institute, 20132 Milan, Italy; 12Medical School, Vita-Salute San Raffaele University, 20132 Milan, Italy; 13Neuroscience and Human Genetics Department, Meyer Children’s Hospital IRCCS, 50134 Florence, Italy

**Keywords:** arylsulfatase A, lysosomal storage disorder, metachromatic leukodystrophy, newborn screening, tandem mass spectrometry

## Abstract

Metachromatic leukodystrophy (MLD) is a rare inherited disorder of lysosomal storage, caused by a deficiency in the arylsulfatase A (ARSA) enzyme, leading to toxic accumulation of sulfatides, which progressively impair motor and cognitive function. MLD is a candidate for inclusion in newborn screening (NBS) programs, due to the narrow pre-symptomatic window for effective therapeutic intervention. We set up a prospective pilot NBS program for MLD in Tuscany, based on a two-step approach. The first-tier test quantified four sulfatides; if levels exceeded the cut-off, we performed the second-tier test by measuring ARSA activity on the same neonatal dried blood spot (DBS). We performed the first-tier test on 42,262 newborns over two years and the second-tier test on residual neonatal DBS from 90 of them (0.21%). We recalled 10 newborns (0.02%) for an additional DBS, due to insufficient residual material for a second-tier test (n = 4) or to low ARSA activity (n = 6). We found normal ARSA activity in all new DBS and identified no new cases of MLD. Retrospective analysis of eight neonatal and fifteen non-neonatal DBS from patients with genetically confirmed MLD showed that the algorithm accurately identified MLD patients. This diagnostic algorithm proved feasible and accurate for early detection of MLD in prospective NBS.

## 1. Introduction

Metachromatic leukodystrophy (MLD) is a rare inherited disorder of lysosomal storage, affecting approximately 0.16–1.85/100,000 live newborns in Europe [1]. MLD is caused by bi-allelic variants in the *arylsulfatase A (ARSA)* gene, leading to a deficiency in the lysosomal ARSA enzyme, resulting in sulfatide accumulation throughout the body, particularly in myelin-forming cells of the central and peripheral nervous system [2]. Sulfatides are toxic and, in addition to demyelination, neuroinflammation and neurodegeneration, cause a consequent progressive decline in motor and cognitive functions, accompanied by multisystemic complications, ultimately leading to premature death [3,4].

In individuals with ARSA deficiency, symptoms of MLD typically appear after a period of normal physical and cognitive development. Based on the age of symptoms onset, MLD can be classified into distinct phenotypes. Early forms include late-infantile MLD (LI-MLD), with onset before 2.5 years of age, and early-juvenile MLD (EJ-MLD), with onset between 2.5 and 7 years of age. Late-onset forms include late-juvenile MLD (LJ-MLD), with onset between 7 and 16 years of age, and the adult form, with onset after 16 years of age [5,6]. Early disease onset is associated with faster and more aggressive disease progression, [7,8].

Diagnosis of MLD is based on biochemical tests including sulfatide levels and ARSA residual enzyme activity, as well as genetic testing for *ARSA* gene mutations. These tests also allow differential diagnosis from other rare phenotypically and biochemically similar disorders such as prosaposin B deficiency and multiple sulfatase deficiency (MSD) [6].

Late-onset MLD patients, if diagnosed during the early symptomatic phase, can benefit from allogeneic hematopoietic stem cell transplantation (HSCT), in an attempt to replace the defective enzyme.

However, this treatment is generally ineffective once the disease has already become symptomatic, as well as in early-onset forms [9]. In the last few years, atidarsagene autotemcel (arsa-cel)—a gene therapy containing an autologous hematopoietic stem and progenitor cell (HSPC) population transduced ex vivo with a lentiviral vector encoding human ARSA cDNA—has become the standard treatment for LI or EJ-MLD. This therapy has proven high effective in preserving cognitive and motor functions in pre-/early-symptomatic patients [10]. Early access to gene therapy can halt disease progression, completely reversing a devastating prognosis for both the patient and their family. Currently, many patients are diagnosed at a symptomatic stage and are no longer eligible for any effective treatment. With this groundbreaking treatment now available, and the critical need for early intervention to archive optimal clinical outcomes, MLD is a strong candidate disorder for inclusion in newborn screening (NBS) programs.

To date, the first prospective NBS pilot programs for MLD were initiated in Washington (USA) and Germany [11,12], and additional prospective NBS pilot programs for MLD have been started in several other countries [13,14].

A research project enabling the identification of the most efficient sulfatides for MLD detection by using liquid chromatography–tandem mass spectrometry (LC-MS/MS) (unpublished data) was started in Tuscany in June 2017. Based on the result of this project, we set up and initiated a prospective pilot NBS program for MLD with a target of screening 80,000 newborns in the Region. Here, we present the preliminary results of this study, which pursued the multiple objectives of (i) setting up a sequential diagnostic algorithm for MLD screening using neonatal DBS specimens collected between 48–72 h of life; (ii) validating the analytical procedures; (iii) assessing the ad interim results of the prospective pilot NBS program for MLD in Tuscany; (iv) confirming the diagnostic performance of the developed algorithm by a proof-of-concept retrospective analysis of neonatal and non-neonatal DBS specimens from patients with an overt, genetically confirmed MLD diagnosis.

## 2. Materials and Methods

### 2.1. Setting

The study was conducted at the laboratories of the Newborn Screening, Clinical Biochemistry and Clinical Pharmacy, Meyer University Children’s Hospital (Florence, Italy). It received approval from the Institutional Review Boards (IRBs) of the Meyer University Children’s Hospital (Florence, Italy) (Approval n. 125/2022). The study was undertaken in accordance with the principles of Good Clinical Practice and the Declaration of Helsinki.

### 2.2. Samples Collection for the Prospective MLD Newborn Screening Program

We set up a prospective pilot NBS program for MLD in Tuscany, with a target of screening 80,000 newborns for this disease. In line with national guidelines in Italy, neonatal DBS samples are routinely collected within the first 48–72 h of life by heel stick, spotted on filter paper (903^®^, Whatman GmbH, Dasel, Germany), dried and sent by courier to the screening center [15]. On residual DBS samples of 42,262 newborns collected between 13 March 2023 and 28 February 2025, we performed additional analyses for MLD screening, according to the diagnostic algorithm that we developed (Figure 1). Ad hoc written informed consent for the inclusion in this pilot NBS program was obtained from parents prior to sample collection. Less than 0.2% of parents declined participation in the pilot project, corresponding to approximately 40 declinations out of an average of 21,000 births per year.

### 2.3. Chemicals and Instrumentation

We purchased labelled internal standard of D_5_-C16:0-sulfatide (D_5_-C16:0-S) from Revvity (Waltham, MA, USA). We prepared a working solution in methanol at the concentration of 10 nmol/L and stored at −20 °C. We obtained methanol, isopropanol, acetonitrile, water and formic acid, all LC-MS grade, from Biosolve Chimie SARL (Dieuze, France).

We used LC-MS/MS equipment consisting of a Nexera LC20AD XR system, with autosampler, vacuum degasser and column oven, from Shimadzu (Tokyo, Japan), coupled with an API 4500 triple quadrupole from ABSciex (Toronto, ON, Canada) equipped with a Turbo V ESI source. We processed the data using Analyst 1.7.2 and SciexOS 2.0 software (Toronto, ON, Canada) and performed quantitation by comparing the analytes to internal standard (D_5_-C16:0-S) signal ratios.

### 2.4. MLD Newborn Screening Program

We measured the concentrations of four sulfatides (C16:0-S, C16:1-OH-S, C16:0-OH-S, C16:1-S), which were identified in a 2017 research project (unpublished data), in DBS using an LC-MS/MS method. The analytical performance of all sulfatides was evaluated before the pilot study.

We established cut-off values of the four sulfatides at the 99.9th percentile based on 5500 samples from anonymous healthy newborns. Informed consent was not required from families of these, as we treated all samples in a fully anonymous manner and present them only as aggregated data.

Starting 1 November 2024, we modified these cut-offs to the 99.0th percentile of the reference population in order to increase the sensitivity of the first-tier test while assuring high specificity by the second-tier test.

We retested duplicate samples with levels of one or more sulfatides over the defined cut-offs and selected positive cases for testing ARSA enzyme activity (second-tier test) [16]. We recalled newborns confirmed positives (ARSA activity < 20% of the daily controls) to delivery hospitals for recollecting a new DBS to retest both sulfatides and ARSA activity.

If the recall test was positive, we asked the newborn’s family to come to our hospital for clinical evaluation, ARSA activity measurement on leukocytes and *ARSA* gene testing. The laboratory procedures for both the first- and second-tier tests are detailed in the following paragraphs and in Figure 2. On newborns identified as affected by MLD during the prospective MLD screening program, we will perform further biochemical, genetic and clinical studies, after obtaining written informed consent from the parents or legal guardians.

### 2.5. Sulfatide Assay (First-Tier Test)

For each DBS, we punched a 3.2 mm diameter circle (nominally containing 3.4 µL of whole blood [17]) into a well of a 96-well plate, reconstituted with 250 μL of a methanolic solution containing the D_5_-C16:0-S internal standard (at 10 nmol/L concentration). We incubated plates for 4 h at 37 °C with gentle shaking and then centrifuged for 10 min at 1800× *g*. We transferred the supernatant into a new 96-well plate and directly injected into mass spectrometer, with an injection volume of 10 µL. We performed chromatographic analysis by using a Zorbax SB-C18 rapid resolution column (2.1 × 15 mm, Agilent Technologies, Santa Clara, CA, USA) to achieve a rapid separation and to maintain a reasonable run time. The mobile phases were 0.1% formic acid in a mixture of water and acetonitrile (65:35, *v*/*v*; phase A) and a mixture of isopropyl alcohol and acetonitrile (70:30, *v*/*v*) containing 0.1% formic acid (phase B). We performed chromatographic separation through a gradient elution as follows: phase B was increased from 1 to 25% over 0.5 min, then to 60% over the next 0.75 min and finally to 100% over 1.5 min. This 100% phase B was maintained for 1 min. Initial conditions were re-established in 0.1 min and held for 1 min. The flow rate was set at 0.6 mL/min, and the column oven temperature was maintained at 50 °C. The total run time, including a brief column re-equilibration period, was 3.5 min.

The instrument collected MRM data in negative mode. We set ion spray voltage at −4500 volts, source temperature at 500 °C, ion source gas 1 and 2 at 45 arbitrary units. The selected MRM transitions and MS tuning parameters are reported in Table A1.

### 2.6. ARSA Activity Assay (Second-Tier Test)

We measured ARSA activity by modifying the method proposed by Hong et al. [16]. We punched 3.2 mm diameter blood spots from each DBS sample into 96-well plates. A blank filter paper spot was included in each plate to detect background activity. In addition, we punched three unknown neonatal controls from the same shipping package to monitor potential degradation due to transport and storage conditions. We added 50 μL of extraction buffer to each DBS punch (patient samples were in duplicates), sealed the plate and shaken on an orbital shaker at room temperature for 4 h. After incubation, we performed an enzyme purification using ultrafiltration spin columns from a commercially available nucleic acid extraction kit (Millipore^®^, Merck KGaA, Darmstadt, Germany). We transferred 30 μL of sample on the top of the silica membrane, followed by the adding of 100 μL of assay buffer [16]. We spun the tubes in a table-top centrifuge at 12,300 rpm (~13,400× *g*) for 10 min, discarded the salts and solvents and kept the purified sample trapped into the filter. To reconstitute it, we turned the filter upside down, added 30 μL of assay buffer to each spin column and spun down again at 6600 rpm (~4000× *g*) for 10 min. We repeated the latter step twice, recollected the purified sample and transferred it into a 96-well plate. Then, we added 10 μL of assay cocktail in each well, capped it and incubated at 37 °C for 16 h (overnight) in an orbital shaker at 250 rpm.

### 2.7. Patients Enrollment for the Proof-of-Concept Study

For the retrospective validation analysis on MLD subjects, eight genetically confirmed pediatric MLD patients evaluated for treatment eligibility at San Raffaele Hospital—the Italian qualified treatment center for MLD gene therapy—were included in this analysis. Informed consent for the use of biological material was obtained at the clinical site. For these patients, their initial neonatal DBS samples were retrieved from the respective regional NBS laboratory following their diagnosis (2–10 years after collection). Additionally, the Voa Voa! Amici di Sofia–Onlus patient advocacy organization provided non-neonatal DBS specimens from 15 other MLD patients (aged 2–35 years at the time of DBS collection). Written informed consent from parents was obtained for all MLD patients enrolled in the retrospective phase.

On the eight neonatal DBS samples from patients with confirmed MLD, we performed the first-tier test according to our MLD newborn screening algorithm, to assess whether they would have been correctly identified as screen-positive at first-tier test, had NBS been implemented at the time of birth. However, as neonatal DBS samples had been collected several years before this analysis, we could not measure ARSA enzyme activity due to the natural degradation of the enzyme at room temperature (i.e., the routine storage condition for DBS specimens).

For non-neonatal samples, we established age-specific cut-off values according to the population size from different age classes (Table A3). We calculated these values considering a volume of 3.4 µL of whole blood for each punch.

### 2.8. Validation of the Analytical Procedures

#### 2.8.1. Accuracy and Precision

For testing the accuracy of the proposed NBS method, we used external reference material (three QC levels for C16:0-S) provided by Archimed Life (Vienna, Austria). We calculated accuracy through comparison of the measured results with expected values.

To assess assay reproducibility, we determined inter-run precision by measuring a real pooled sample over ten runs on ten different days. We used the resulting data to calculate the mean value and the relative standard deviation (RSD) as 100 × (total standard deviation)/mean. Our goal was to have an RSD of ≤15% for each analyte. We used the same real pooled sample to determine the within-run precision by analyzing the pooled sample ten times in the same batch. Again, we calculated the mean value and RSD.

#### 2.8.2. Carryover and Spot Homogeneity

We investigated carryover by injecting 20 μL of extraction solvent (i.e., a volume double that used in NBS routine) after 10 samples for three times and monitored the response at the retention times of the four different sulfatides. We also verified the spot homogeneity by punching at different locations, including left, right, center, top and bottom edge of three different spots at three different concentrations.

#### 2.8.3. Stability

We evaluated the short-term stability study on DBS under different storage temperatures, including −20 °C, +4 °C, room temperature and +37 °C, up to four weeks. We examined the long-term stability over a period of 24 months at −20 °C.

## 3. Results

### 3.1. Reference Values for First-Tier Test

We first set up the reference values of the four sulfatides and their sum, based on the analysis of 5500 samples from healthy newborns and calculated as the 99.9th percentile. The reference values resulted as C16:0-S < 196 nmol/L, C16:0-OH-S < 228 nmol/L, C16:1-S < 41 nmol/L, C16:1-OH-S < 60 nmol/L, sum of the four sulfatides < 485 nmol/L.

Starting 1 November 2024, we modified these cut-offs to increase sensitivity and set as the 99.0th percentile, resulting as C16:0-S < 161 nmol/L, C16:0-OH-S < 182 nmol/L, C16:1-S < 33 nmol/L, C16:1-OH-S < 50 nmol/L, sum of the four sulfatides < 398 nmol/L.

Based on these cut-offs, we retested samples with sulfatide values above the cut-off in duplicates, and we selected positive cases for second-tier testing.

### 3.2. Validation of the Analytical Procedures

Table A2 reports the variations of intra-run and inter-run determinations on DBS. The method displayed a good reproducibility, accuracy and precision for the quantification of the four sulfatides.

Regarding carry-over, we found that the signals following the injection of 20 μL of extraction solvent after a series of 10 samples were always absent. We found that lateral punching of the DBS samples was associated with a non-significant increase (3 to 5%) in the measured levels of the four analytes when compared to central punching.

All analytes in the DBS matrix were stable at −20 °C, +4 °C, room temperature and +37 °C for up to four weeks and at −20 °C for up to 24 months, with a CV% < 15% for all tested temperatures (Figure A1).

### 3.3. Ad Interim Analysis from the Prospective Pilot

We included 42,262 newborns in the prospective MLD NBS program and performed first-tier tests on their residual neonatal DBS samples (Figure 3).

Figure A2 reports on a comparison of sulfatide profile in negative control and a MLD newborn.

We identified 90 samples (0.21%) with levels of one or more sulfatides over the defined cut-off values in the first-tier test and performed the second-tier test on all of them.

At second-tier analysis, we found ARSA enzyme activity within the defined cut-off values for 80 out of 90 newborns (screen negative subjects). Figure 4 reports an example of an ARSA enzyme activity test on a negative control newborn and on a MLD positive case.

The remaining six/ten newborns presented low ARSA activity (<20% of the daily controls); thus, we recalled them to provide a new DBS sample. We also recalled the other four newborns, as residual material on the neonatal DBS was insufficient to perform a second-tier test (Figure 3). The recall rate resulted as 0.02%.

In all individuals recalled for a new DBS (analyzed in duplicate), we found ARSA activity within normal values, and we identified no MLD true positive cases. For the first newborn who tested positive at the beginning of the pilot, we simultaneously performed both biochemical and genetic tests (*ARSA* gene analysis), in agreement with family. The sum of the sulfatides was slightly abnormal, while both ARSA activity and the *ARSA* gene were normal. We classified the newborn as a false positive.

Since then, we changed the routine diagnostic flow-chart, and we now perform genetic testing only in the case of confirmed ARSA enzymatic deficiency in both samples (neonatal DBS and recall DBS).

### 3.4. Proof-of-Concept Analysis on Patients with Genetically Confirmed MLD

We retrieved neonatal DBS specimens from eight patients (aged 2–10 years at the time of inclusion in the study) with overt, genetically and biochemically confirmed MLD and measured concentrations of four sulfatides (C16:0-S, C16:1-OH-S, C16:0-OH-S, C16:1-S) (Table 1 and Figure 5).

Based on the diagnostic algorithm we used for the current pilot NBS, all patients would have been correctly identified to be retested and to undergo second-tier test. Namely, we found that the concentrations of each single sulfatide, as well as the sum of the four sulfatides, were above the cut-off values for all patients. We could not proceed with second-tier tests (i.e., measurement of ARSA enzyme activity) on these residual neonatal DBS due to ARSA enzyme degradation, as these specimens had been stored at room temperature (as defined by the national regulation) for 2–10 years after collection.

For non-neonatal samples, we established age-specific cut-off values at the 99.0th percentile of reference populations from different age classes, including 1–3 months (n = 46), 3–6 months (n = 26), 6–12 months (n = 54), 1–3 years (n = 142), 3–18 years (n = 246) and >18 years (n = 333). However, the cut-off values of the four sulfatides were almost comparable across the age classes from 1 year to adulthood, and we therefore merged them in a unique age class (>1 year). We report cut-off values in Table A3.

We also analyzed non-neonatal DBS samples from an additional 15 patients with genetically confirmed MLD (aged 2 to 35 years at the time of DBS collection). All patients, except one (patient n° 15 in Table 2), presented with an early-onset phenotype. Patient n°15 was diagnosed with the late-juvenile form. Table 2 reports their concentrations of C16:0-S, C16:0-OH-S, C16:1-S and C16:0-OH-S and of the sum of the four sulfatides. All patients had concentrations of C16:0-OH-S and C16:1-OH-S or the sum of the four sulfatides above the cut-off values, established from an age-matched population. Three patients showed C16:0-S levels within the cutoff range, while 14 out of 15 patients had increased levels of C16:1-S.

Enzymatic activity tests conducted on DBS from all non-neonatal subjects showed undetectable product formation, including the subject with late-juvenile onset, who was in an advanced stage of the disease (Figure 5F).

## 4. Discussion

Anticipating an MLD diagnosis by NBS has the potential to revolutionize the clinical course of the disorder, making it possible to identify affected patients within the critical window for effective therapy with arsa-cel (in early-onset forms) or HSCT (in late-onset forms). The benefits of identifying MLD early in the pre-symptomatic phase has already been demonstrated, as reported in the trial on arsa-cel, where most pre-symptomatically treated children were diagnosed due to an already affected sibling [10].

In this study, we set up and implemented an NBS strategy, based on a first-tier test (measurement of sulfatide concentrations on DBS by LC-MS/MS), second-tier test (ARSA residual enzyme activity on DBS), recalling infants to delivery hospitals to retest sulfatides, ARSA activity and genetic sequencing of the *ARSA* gene. We fully validated the laboratory procedures.

The prospective pilot program is still ongoing and has a target of 80,000 newborns to be screened for MLD. Based on the ad interim analysis on a first cohort of 42,262 newborns, we showed that this NBS algorithm is feasible, as the false positive rate of the first-tier test was 0.21%.

Our first-tier test yielded a false positive rate consistent with that reported by Wu et al. [13] for the diagnostic algorithm developed for MLD NBS in the UK, which, however, was based solely on the quantification of C16:0-S as the first-tier marker (0.3%). Moreover, our rate was lower than that reported for the US algorithm, which is also based on C16:0-S quantification as a first-tier test (0.71%) [11]. Conversely, in the same study, a lower false positive rate (0.13%) was reported when ARSA enzyme activity was used as the first-tier test, with C16:0-S quantification applied as a second-tier test. In a German study, the use of C16:1-OH-S, either alone or in combination with C16:0-S, as a first-tier test further reduced the false positive rate to approximately 0.05%, with the addition of second-tier ARSA activity testing bringing the rate close to zero [18].

In our study, the recall rate was also low (0.02%), thus minimizing potential psychosocial consequences for the families. Of note, four out of ten cases in which we needed to recall the subject were due to the lack of sufficient residual material to perform accurate measures of residual ARSA enzyme activity on the DBS.

A few days after beginning the pilot project, we found the first abnormal result in both the first- and second-tier test. According to our initial diagnostic algorithm, we asked for a second DBS sample, confirming a slight alteration in the sulfatide profile but normal ARSA activity and *ARSA* gene analysis.

To date, we found no true positive MLD cases in the screening of 42,262 newborns. While this reflects the rarity of MLD, it also limits to evaluate the real-world sensitivity and positive predictive value of the diagnostic algorithm.

However, in a retrospective proof-of-concept analysis of neonatal DBS from eight patients with overt, genetically confirmed MLD, we showed that these cases would have been correctly identified as screen-positive by the first-tier test. Notably, the true concentrations of these sulfatides at the time of neonatal DBS collection could have been even higher than those retrospectively measured in this study, due to the potential degradation of the sample over time.

The impossibility of measuring residual ARSA enzyme activity in retrospectively retrieved neonatal DBS specimens, due to the natural degradation of the enzyme after storage at room temperature, prevented us from progressing to the subsequent diagnostic steps of the algorithm. Thus, the performance of the full diagnostic algorithm in prospectively identifying new MLD positive cases remains theoretical to date. Nevertheless, it is highly probable that, nowadays, these patients would have been correctly identified as affected by MLD at birth, allowing the start of arsa-cell therapy in the pre-symptomatic phase, determining a different disease course [12].

The cost of the first-tier screening test using a mass spectrometry-based assay is relatively low when integrated into an existing NBS platform; based on our project estimates, these additional costs amount to approximately €5 per test, considering reagents, instrument servicing, consumables and laboratory personnel. Moreover, while follow-up procedures such as enzyme assays, genetic testing and neuroimaging may increase healthcare costs, the diagnostic algorithm based on a multi-tier approach has been shown to result in a low false positive rate, thereby reducing the burden of unnecessary follow-up. On the other hand, MLD is a progressive and devastating neurodegenerative disease with a high cost burden when diagnosed after symptom onset. Early treatment can reduce the need for intensive medical care, long-term hospitalization, mobility aids and specialized education, leading to significant long-term savings for healthcare systems and families, as demonstrated in studies on the economic implications of MLD NBS [19].

Based on the first-tier test performed on these retrospectively collected neonatal DBS samples, it emerged that C16:0-S and C16:1-OH-S, as well as the sum of the four sulfatides, could be considered as primary markers of neonatal screening for MLD, as they were above the cut-off values for all patients, as elsewhere reported [12,18]. Moreover, we found that C16:0-OH-S and C16:1-S were also above the cut-off levels and should be considered noteworthy as additional primary markers.

With the use of the sulfatide profile as a proof-of-concept analysis on non-neonatal DBS samples from 15 MLD patients, we further showed that concentrations of C16:0-OH-S and/or C16:1-OH-S and/or C16:1-S (as well as C16:0-S in 12/15) were remarkably increased in all samples, further suggesting the potential role of the DBS sulfatide profile as a primary marker for MLD biochemical diagnostic purposes.

Fourteen out of fifteen non-neonatal DBS samples were from early-onset MLD patients, with only one from a late-juvenile case. While this is not surprising due to the higher prevalence of early-onset forms [1], we could not evaluate the diagnostic performance of the algorithm in detecting late-onset MLD nor analyze the role of the four sulfatides in differentiating disease phenotypes. Additional studies on late-onset MLD cohorts are needed to assess the performance of this diagnostic algorithm on this disease phenotype. Notably, although late-juvenile MLD cases are not currently eligible for arsa-cel, a phase III clinical trial is underway to assess the efficacy and safety of gene therapy in this disease phenotype [NCT04283227]. Pending the results of this trial, early detection of late-juvenile MLD through NBS has the potential to revolutionize the disease course for these patients. Further studies on freshly collected neonatal DBS from early- and late-onset MLD patients are required to assess that role. When screen-positive patients are found by biochemical tests, differential diagnosis with other MLD-mimicking disorders is required, including prosaposin B deficiency due to bi-allelic variants in *prosaposine* gene (*PSAP*) and multiple sulfatase deficiency (MSD) due to bi-allelic variants in the *sulfatase modifying factor 1* gene (*SUMF1*). Genetic sequencing of *ARSA*, *SUMF1* and *PSAP* genes is required for differential diagnosis. According to our algorithm, we performed only *ARSA* gene sequencing in patients testing positive to the second-line test, as the IRB that approved this study considered as inappropriate the inclusion of genetic investigation of prosaposin B deficiency or MSD. Indeed, these two diseases do not meet the Wilson and Jungner criteria for inclusion in NBS [20] or their revisitation [21], as, to date, there is no treatment available for them. Early recognition would have a limited impact on their clinical course, and it is almost impossible to establish the overall benefits of screening over potential harm. In routine clinical practice outside the aims of this pilot, genetic sequencing of *SUMF1* and *PSAP* genes can be performed based on clinical judgment, after obtaining an ad hoc written informed consent from the parents or legal representatives.

The pilot program presented in this study is still ongoing to meet the final target of screening 80,000 newborns for MLD. Considering the low proportion of positive cases coming by the first-tier test, we deemed it reasonable to modify the cut-off values for the first-tier test, considering the 99.0th percentile (instead of using the 99.9th percentile) of the 5500 samples from anonymous healthy newborns. Starting November 2024, we implemented the new cut-offs in the NBS, to potentially increase the sensitivity of the first-tier test. This choice inevitably increased the percentage of DBS samples undergoing the 2TT, rising from 0.21% with the previous cut-offs to 0.72% with the current cut-offs. Notably, this second-tier test rate is similar to that reported for the US diagnostic algorithm based on the first-tier C16:0-S assay (0.71%) [13]. According to our algorithm, specificity is guaranteed by the second-tier test, and no additional recall procedures occurred as a result of this modification of the reference values. Setting up the cut-off value to 99.0th percentile strongly improves the diagnostic sensitivity of both C16:OH-S and C16:0-S sulfatides (eight/eight patients identified with new values; see Table 1).

A multicenter implementation of the proposed diagnostic algorithm in national and international NBS programs is recommended to confirm its feasibility and accuracy across different populations and operational settings.

## 5. Conclusions

MLD is a life-threatening progressive disease with a limited pre-symptomatic therapeutic window to effectively start gene therapy and obtain an optimal clinical outcome. Thus, pre-symptomatic diagnosis of MLD through preventative health programs such as NBS is essential.

In this study, we report a diagnostic algorithm for NBS MLD screening, demonstrating its validity and feasibility in prospective NBS programs. We also confirmed its accuracy in retrospectively identifying patients with symptomatic, genetically confirmed MLD.

As of today, no new case of MLD was identified within this prospective NBS program, likely due to the rarity of the disease. Thus, the real-world sensitivity and positive predictive value of this diagnostic algorithm cannot yet be estimated. Nevertheless, based on the residual neonatal DBS analysis of genetically confirmed MLD positive cases, we are confident this algorithm will enable early detection and prompt referral to the competent clinical unit of any future screen-positive patient, allowing pre-symptomatic treatment initiation and a significantly improved clinical outcome.

## Figures and Tables

**Figure 1 IJNS-11-00030-f001:**
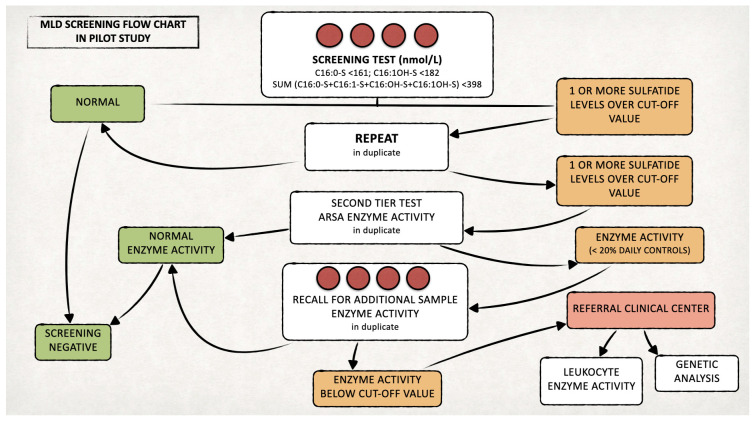
Proposed algorithm for the MLD newborn screening.

**Figure 2 IJNS-11-00030-f002:**
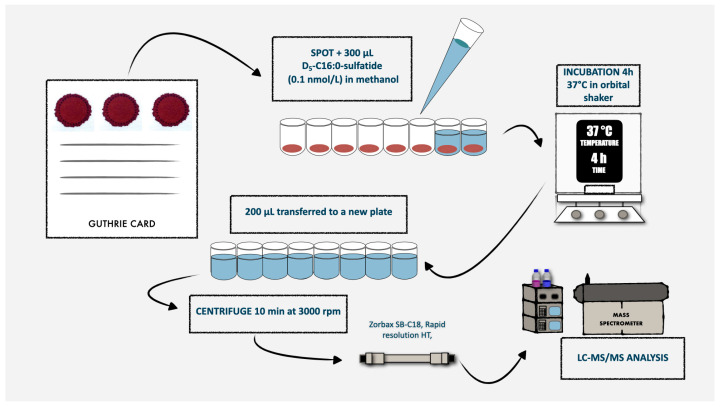
Schematic representation of the routine laboratory procedures for the first-tier test used in MLD newborn screening.

**Figure 3 IJNS-11-00030-f003:**
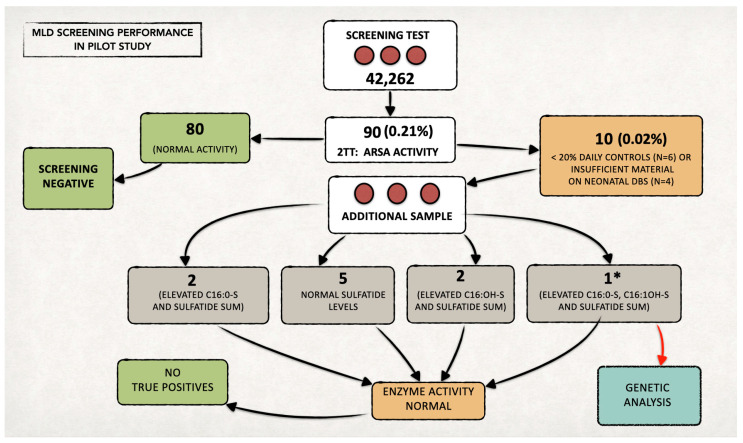
Overall data from the MLD newborn screening program in Tuscany over two years. We performed the second-tier test (2TT) on 90 newborns (0.21% of the screened neonatal population). The recall rate was approximately 0.02% (n = 10). * In one case, since all the quantified sulfatides also remained increased on the second sample (analyzed in duplicate), we proceeded with the genetic study, in agreement with the family. No genetic alterations were highlighted, and the subject was classified as a false positive. Since then, the diagnostic flow-chart has been changed, and genetic testing is now performed only in the case of ARSA enzymatic deficiency in both samples (neonatal DBS and second DBS after recall).

**Figure 4 IJNS-11-00030-f004:**
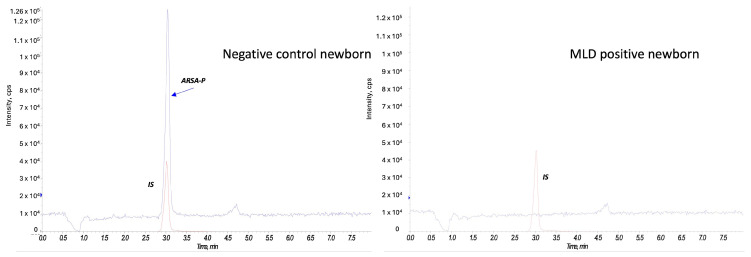
ARSA enzyme activity in a negative control newborn and a MLD case. The red peak represents the internal standard (IS), and the blue peak corresponds to the product of the ARSA activity (ARSA-P) in a negative control newborn (on the left) and in a MLD patient (on the right).

**Figure 5 IJNS-11-00030-f005:**
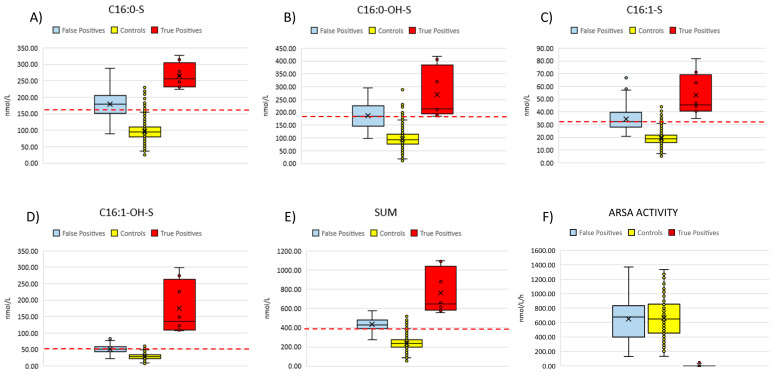
Summarizes the distribution of the four sulfatides (**A**–**D**) and their sum (**E**) in the neonatal DBS from 90 false positive cases, eight confirmed MLD patients and controls (n = 3154). (**F**) compares ARSA activity measured in fresh DBS samples from false positive cases (n = 90), non-neonatal MLD patients (n = 15) and controls (n = 120). Red dashed lines represent the current cut-off values for each of the four sulfatides and the sum.

**Table 1 IJNS-11-00030-t001:** Concentrations of the four sulfatides (C16:0-S, C16:0-OH-S, C16:1-S and C16:0-OH-S) and their sum, measured in the residual neonatal DBS samples, stored at room temperature, from eight patients with overt, genetically confirmed MLD.

Patient ID	Year of Birth	C16:0-S (n.v. < 161 nmol/L)	C16:0-OH-S (n.v. < 182 nmol/L)	C16:1-S (n.v. < 33 nmol/L)	C16:1-OH-S (n.v. < 50 nmol/L)	Sum (n.v. < 398 nmol/L)
1	2022	324	419	64	307	1114
2	2018	247	214	41	113	614
3	2017	232	210	41	148	631
4	2022	280	215	45	122	662
5	2016	265	319	71	225	880
6	2014	328	406	82	275	1091
7	2021	234	186	47	108	575
8	2020	225	190	35	107	556

The reported cut-offs have been in use since November 2024.

**Table 2 IJNS-11-00030-t002:** Concentrations of the four sulfatides (C16:0-S, C16:0-OH-S, C16:1-S and C16:0-OH-S) and their sum, measured by LC-MS/MS in the non-neonatal DBS samples from 15 patients with overt, genetically confirmed MLD.

Patient ID	Age at DBS Collection (Years)	Sex	C16:0-S (n.v. < 413 nmol/L)	C16:0-OH-S (n.v. < 428 nmol/L)	C16:1-S (n.v. < 57 nmol/L)	C16:1-OH-S (n.v. < 115 nmol/L)	Sum (n.v. < 911 nmol/L)
1	2	F	1483	2513	232	1224	5452
2	2	F	707	1160	133	616	2616
3	3	F	513	1038	88	410	2050
4	3	F	1294	2136	227	1099	4756
5	6	M	806	1060	136	424	2426
6	7	F	554	752	86	274	1665
7	7	F	380	700	75	296	1451
8	7	F	397	670	70	295	1433
9	8	M	417	602	58	156	1233
10	11	F	376	701	50	197	1324
11	15	M	618	1330	100	471	2520
12	16	M	520	616	181	453	1770
13	16	M	573	897	91	409	1970
14	19	F	437	633	73	258	1400
15	35	M	805	1222	113	385	2526

The reported cut-offs are set at the 99.0th percentile of a reference population aged > 1 year (n = 721).

## Data Availability

Additional data supporting reported results can be made available upon written request to the corresponding author.

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
