# Peer review of "Newborn Screening for Metachromatic Leukodystrophy in Tuscany: The Paradigm of a Successful Preventive Medicine Program"

_2409-515X, 2025, doi:10.3390/ijns11020030_

Round 1

Reviewer 1 Report

Comments and Suggestions for Authors

Please see attachments for review comments

Author Response

R#1

The authors have submitted a well written and interesting paper that will make an important

contribution to the NBS society. There are however some minor considerations. […] In summary, the authors provide important novel newborn screening data in a well written manner. However, valuable data are left out and should be included, suggested in the recommendations above.

Reply: We truly thank the Reviewer for the positive feedback and valuable suggestions. We now implemented the manuscript, addressing all Reviewer’s suggestion.

Title: Please consider adding Country in the title to clarify where the pilot program is performed.

Reply: As suggested, we modified the title as follows:

Newborn Screening for Metachromatic Leukodystrophy in Tuscany: The Paradigm of a Successful Preventive Medicine Program

Introduction:

Line 89; Please consider mentioning the published pilot studies for MLD. Hong et al 2021, 27000

Newborn screened in Washington, Laugwits et al 2024, 109 000 newborns screened in Germany

and the UK pre-pilot perhaps with 3700 screened.

Reply: As suggested, we now implemented the bibliography of the manuscript by adding the suggested papers.

Line 94; “data not shown”, please consider changing to “unpublished data” it that is the case or

provide a reference if published.

Reply: As suggested, we replaced it with “unpublished data”.

Materials and Methods:

Line106-111; Please provide information of how many children are born in the region and

screened in the current NBS program, acceptance and participation to the program in the

population.

Reply: As suggested, we now added this information at the end of paragraph 2.2, as follows:

Less than 0.2% of parents declined participation in the pilot project, corresponding to approximately 40 declinations out of an average of 21,000 births per year.

Line 120; Please clarify if the written consent is for the MLD pilot or if it’s a routine in the current

NBS program.

Reply: We now clarified that the written consent was an ad hoc form, specific for this pilot. Indeed, the routine NBS in Tuscany does not require the obtainment of a written informed consent (but rather of a written informed dissent, if pertinent).

Line 137; Is the 2017 research program unpublished data? If so, clarify or provide a reference for

the project.

Reply: We now clarified in the manuscript that these data are unpublished.

Line 160; Please provide a reference for the estimation of a 3.2mm DBS containing 3.4uL blood.

Reply: As suggested, we now added a reference for this statement.

Results:

For the 42,262 screened newborns, no data is provided other than the number of samples that

underwent 2ndtier and the number of newborns that were recalled. Please provide, preferably,

graphical data for sulfatide distribution of screen negative versus screen positive newborns and

for the 90 newborns that had elevated sulfatides, please provide the same comparison in %

enzyme activity negative newborns versus referrals.

Reply: We truly thank the Reviewer for the suggestion. We now added a figure (Figure 5) summarizing the distribution of the four sulfatides and their sum, in the neonatal DBS from the 90 false positive cases, the eight true MLD patients, and controls. Also, the figure shows ARSA activity assessed in fresh DBS samples from false positive cases (n=90), non-neonatal MLD patients (n=15), and controls (n=120).

For the non-neonatal (15) patient samples, sulfatide concentrations are presented in Table 2.

The authors do not include enzymatic activity data for them, please provide the data in table of

graph, or if no enzymatic activity of ARSA was performed, please state so.

Reply: We thank the Reviewer for this comment. ARSA activity was assessed also in non-neonatal samples, but levels of product formation resulted undetectable in all cases (including the patient with late juvenile onset MLD). We now added this information in the text, at the end of paragraph 3.4:

Enzymatic activity tests conducted on DBS from all non-neonatal subjects showed undetectable product formation, including the subject with late juvenile onset, who was in an advanced stage of the disease (Figure 5, panel F)”.

Reviewer 2 Report

Comments and Suggestions for Authors

A very well presented paper with well thought through design, collected a vast amount of lab data that demonstrated test cutoff values were challenged then lowered to improve test sensitivity for MLD, overall a well structured pilot study.  The Tuscany MLD pilot study is one of the few large scale newborn screening pilot studies for MLD, contributing real world evidence to inform policy makers in newborn screening.  I look forward to read the final report when the 80,000 sample analyses are complete.

I suggest the following points for consideration as minor changes:

  1. In the abstract the term 'false positive' was used for 6 out of 10 samples.  This referred to recalling the baby for a second NBS bloodspot to clarify the results in the original sample.  The recall for a 2nd bloodspot was actually developed into the full screening algorithm (Figure 1) so calling these 6 results as false positives was a strong description and doesn't give the algorithm justice.  Since they were not the final result from the full algorithm which determined all 6 samples as screen negatives.  Clarification for readers in the abstract would be helpful.  Could authors comment if the ARSA stability in the initial bloodspot the cause of the initial low enzyme result.   
  2. section 3.2  line 256 "...extraction solvent after a series of 10 samples, were ever absent."  may be better worded as 'were always absent'.  
  3. Figure 3 Caption, the description "altered" and "alterations" of sulphatide results, perhaps best to say they remained increased for clarity.
  4. Conclusion line 372.  The C16:1-OH S was also reported to be increased in MLD positive newborn bloodspots in a multi-centre report, and should be referenced - Bekri  et al  Higher precision, first tier newborn screening for metachromatic leukodystrophy using 16:1-OH-sulphatide. Genet Metal 2024 May;142(1):108436

Author Response

R#2

A very well presented paper with well thought through design, collected a vast amount of lab data that demonstrated test cutoff values were challenged then lowered to improve test sensitivity for MLD, overall a well structured pilot study.  The Tuscany MLD pilot study is one of the few large scale newborn screening pilot studies for MLD, contributing real world evidence to inform policy makers in newborn screening.  I look forward to read the final report when the 80,000 sample analyses are complete.

Reply: We truly thank the Reviewer for the positive feedback and valuable suggestions. We now implemented the manuscript, addressing all Reviewer’s suggestion.

I suggest the following points for consideration as minor changes:

In the abstract the term 'false positive' was used for 6 out of 10 samples.  This referred to recalling the baby for a second NBS bloodspot to clarify the results in the original sample.  The recall for a 2nd bloodspot was actually developed into the full screening algorithm (Figure 1) so calling these 6 results as false positives was a strong description and doesn't give the algorithm justice.  Since they were not the final result from the full algorithm which determined all 6 samples as screen negatives.  Clarification for readers in the abstract would be helpful. Could authors comment if the ARSA stability in the initial bloodspot the cause of the initial low enzyme result.

Reply: We thank the Reviewer for this suggestion. We now revised the abstract, by clearly stating that:

 “We recalled 10 newborns (0.02%) for an additional DBS, due to insufficient residual material for second-tier test (n=4) or to low ARSA activity (n=6). We found normal ARSA activity in all new DBS, and identified no new cases of MLD.”.

Regarding the possible cause for low ARSA activity in the false positive cases, we exclude that this was related to enzyme stability on the initial neonatal DBS samples, as they are routinely processed within one week from collection. Moreover, each analytical session includes the analysis of an intra-batch sample, stored at the same conditions and for the same period of the index sample.

The most probable cause for low ARSA activity results, in our 6 false positive cases, is the loss of the enzyme during sample preparation, which includes a critical filtration step. Notably, ARSA activity was always assessed in duplicate, and, according to our protocol, the test is repeated in case of discrepant results.     

section 3.2 line 256 "...extraction solvent after a series of 10 samples, were ever absent."  may be better worded as 'were always absent'. 

Reply: We revised the sentence as suggested.

Figure 3 Caption, the description "altered" and "alterations" of sulphatide results, perhaps best to say they remained increased for clarity.

Reply: We revised the sentence as suggested.

Conclusion line 372.  The C16:1-OH S was also reported to be increased in MLD positive newborn bloodspots in a multi-centre report, and should be referenced - Bekri  et al  Higher precision, first tier newborn screening for metachromatic leukodystrophy using 16:1-OH-sulphatide. Genet Metal 2024 May;142(1):108436

Reply: We now acknowledged this paper in the discussion (new ref 18), as suggested.

Reviewer 3 Report

Comments and Suggestions for Authors

With the advent of gene therapy for MLD and the critical importance of early intervention, the study addresses a pressing clinical need for early diagnosis through NBS.

This paper presents the design, implementation, and interim results of a prospective pilot newborn screening (NBS) program for Metachromatic Leukodystrophy (MLD) in Tuscany, Italy. The program uses a two-tiered approach: initial quantification of four sulfatides by LC-MS/MS in dried blood spots (DBS), followed by measurement of arylsulfatase A (ARSA) enzyme activity for samples above the sulfatide cutoff. The pilot screened 42,262 newborns over two years, with no new MLD cases identified. Retrospective analysis of DBS from confirmed MLD patients validated the algorithm's ability to detect MLD biochemically. The study also describes analytical validation, including precision, accuracy, and stability testing of the assays.

It is is well written paper but there are multiple limitations that I would like to bring to the authors’ attention;

-The pilot, at the interim analysis, identified no new MLD cases among 42,262 newborns. While this reflects the rarity of MLD, it limits the ability to assess the real-world sensitivity, specificity, and positive predictive value (PPV) of the screening algorithm prospectively. The absence of true positives means the algorithm's performance in a true screening setting remains theoretical.

-The retrospective validation relied on stored neonatal DBS samples from confirmed MLD cases, but ARSA activity could not be measured due to enzyme degradation over time. Therefore, only the first-tier sulfatide screen was tested retrospectively, not the full diagnostic algorithm.

-The retrospective cohort was small (8 neonatal and 15 non-neonatal MLD cases), and most non-neonatal samples were from early-onset cases, limiting insights into the algorithm's ability to detect late-onset MLD or differentiate among phenotypes.

-The paper does not directly compare its algorithm's performance with other published or ongoing MLD NBS programs, such as those in Germany or the US, beyond referencing their existence. A comparative analysis would contextualize the strengths and potential gaps of the Tuscany approach. I strongly urge the authors consider including direct comparison of assay performance, recall rates, and workflow with other published MLD NBS programs.

-The modification of sulfatide cut-offs from the 99.9th to 99.0th percentile to increase sensitivity is reasonable, but the impact of this change on recall rates, specificity, and laboratory workload is not fully detailed in the interim results. The authors predict no additional recalls, but this should be confirmed with further data if possible.

-The study is regionally focused (Tuscany) and may not account for population genetic variability or operational differences in other settings. Broader implementation would require multicenter validation.

-While the clinical necessity of early detection is emphasized, the paper does not address the cost-effectiveness of adding MLD to NBS panels, which is a key consideration for policy decisions. The authors should Incorporate a discussion or analysis of the economic implications of MLD NBS, including laboratory costs, follow-up, and potential savings from early intervention.

- The authors should discuss the impact of diagnosing the late-onset form of MLD via newborn screening, which would necessitate lifelong monitoring, particularly given the limited treatment options currently available.

Author Response

R#3

With the advent of gene therapy for MLD and the critical importance of early intervention, the study addresses a pressing clinical need for early diagnosis through NBS.

This paper presents the design, implementation, and interim results of a prospective pilot newborn screening (NBS) program for Metachromatic Leukodystrophy (MLD) in Tuscany, Italy. The program uses a two-tiered approach: initial quantification of four sulfatides by LC-MS/MS in dried blood spots (DBS), followed by measurement of arylsulfatase A (ARSA) enzyme activity for samples above the sulfatide cutoff. The pilot screened 42,262 newborns over two years, with no new MLD cases identified. Retrospective analysis of DBS from confirmed MLD patients validated the algorithm's ability to detect MLD biochemically. The study also describes analytical validation, including precision, accuracy, and stability testing of the assays. It is is well written paper but there are multiple limitations that I would like to bring to the authors’ attention.

Reply: We truly thank the Reviewer for the positive feedback and valuable suggestions. We now implemented the manuscript, addressing all Reviewer’s suggestion.

-The pilot, at the interim analysis, identified no new MLD cases among 42,262 newborns. While this reflects the rarity of MLD, it limits the ability to assess the real-world sensitivity, specificity, and positive predictive value (PPV) of the screening algorithm prospectively. The absence of true positives means the algorithm's performance in a true screening setting remains theoretical.

Reply: We fully agree with the Reviewer in recognizing that the lack of true positive MLD cases detected with this algorithm represents a limitation of this study; we now better addressed it in the Discussion and in the Conclusions, as follows:

Discussion (page 12, lines 426-428): “To date, we found no true positive MLD cases in the screening of 42,262 newborns. While this reflects the rarity of MLD, it also limits to evaluate the real-world sensitivity and positive predictive value of the diagnostic algorithm.”

Conclusions: “As of today, no new case of MLD was identified within this prospective NBS program, likely due to the rarity of this disease. Thus, the real-world sensitivity and positive predictive value of this diagnostic algorithm cannot yet be estimated. Nevertheless, based on the residual neonatal DBS analysis of genetically confirmed MLD positive cases, we are confident this algorithm will enable early detection and prompt referral to the competent clinical unit of any future screen positive patient, allowing pre-symptomatic treatment initiation and a significantly improved clinical outcome”.

-The retrospective validation relied on stored neonatal DBS samples from confirmed MLD cases, but ARSA activity could not be measured due to enzyme degradation over time. Therefore, only the first-tier sulfatide screen was tested retrospectively, not the full diagnostic algorithm.

Reply: We agree with the Reviewer in recognizing that this represents an unavoidable limitation of our study. We now better acknowledged this issue in the Discussion, as follows:

The impossibility of measuring residual ARSA enzyme activity in retrospectively retrieved neonatal DBS specimens, due to the natural degradation of the enzyme after storage at room temperature, prevented us from progressing to the subsequent diagnostic steps of the algorithm. Thus, the performance of the full diagnostic algorithm in prospectively identifying new MLD positive cases remains theoretical to date.” (page 12, lines 435-442).

-The retrospective cohort was small (8 neonatal and 15 non-neonatal MLD cases), and most non-neonatal samples were from early-onset cases, limiting insights into the algorithm's ability to detect late-onset MLD or differentiate among phenotypes.

Reply: As correctly stated by the Reviewer, this aspect is another major limitation of our study. However, it is worth mentioning that, given the rareness of this disease, this study analysed a remarkable number of neonatal and non-neonatal MLD samples. Moreover, given that early-onset forms represent around 80% of all MLD cases [Chang, Orphanet Journal of Rare Diseases, 2024], it is not surprising that we included only one late juvenile patient. 

We now clearly stated it in the Discussion, as follows:

Fourteen out of 15 non-neonatal DBS samples were from early-onset MLD patients with only one from a late juvenile case. While this is not surprising due to the higher prevalence of early-onset forms [1], we were unable to evaluate the diagnostic performance of the algorithm in detecting late-onset MLD, or to analyse the role of the four sulfatides in differentiating disease phenotypes.” (page 13, lines 469-473).

-The paper does not directly compare its algorithm's performance with other published or ongoing MLD NBS programs, such as those in Germany or the US, beyond referencing their existence. A comparative analysis would contextualize the strengths and potential gaps of the Tuscany approach. I strongly urge the authors consider including direct comparison of assay performance, recall rates, and workflow with other published MLD NBS programs.

Reply: As suggested, we now compared the workflow and performance of our diagnostic algorithm with those used in other pilot NBS programs, in the USA, UK and Germany.

“The prospective pilot program is still ongoing and has a target of 80,000 newborns to be screened for MLD. Based on the ad interim analysis on a first cohort of 42,262 newborns, we showed that this NBS algorithm is feasible, as the false positive rate of the first-tier test was 0.21%. Our first-tier test yielded a false positive rate consistent with that reported by Wu et al. [13] for the diagnostic algorithm developed for MLD NBS in the UK, which, however, was based solely on the quantification of C16:0-S as the first-tier marker (0.3%). Moreover, our rate was lower than that reported for the US algorithm, which is also based on C16:0-S quantification as a first-tier test (0.71%) [11]. Conversely, in the same study, a lower false positive rate (0.13%) was reported when ARSA enzyme activity was used as the first-tier test, with C16:0-S quantification applied as a second-tier test. In a German study, the use of C16:1-OH-S, either alone or in combination with C16:0-S, as a first-tier test further reduced the false positive rate to approximately 0.05%, with the addition of second-tier ARSA activity testing bringing the rate close to zero [18]. (page 12, lines 404-417).

This choice inevitably increased the percentage of DBS samples undergoing 2TT, rising (from 0.21% with the previous cut-offs, to 0.72% with the current cut-offs. Notably, this second-tier test rate is similar to that reported for the US diagnostic algorithm based on first-tier C16:0-S assay (0.71%) [13]. (page 13, lines 503-505).

-The modification of sulfatide cut-offs from the 99.9th to 99.0th percentile to increase sensitivity is reasonable, but the impact of this change on recall rates, specificity, and laboratory workload is not fully detailed in the interim results. The authors predict no additional recalls, but this should be confirmed with further data if possible.

Reply: We thank the Reviewer for the opportunity to address this aspect. We modified the cut-off values of the first-tier test, to increase the sensitivity. This choice inevitably modified the work-flow and increased the percentage of second-tier test on the same neonatal DBS (from 0.21% with the previous cut-offs, to 0.72% with current cut-offs). However, the recall rate was not affected by this choice (0.02%). We now specified this aspect also in the Discussion.

-The study is regionally focused (Tuscany) and may not account for population genetic variability or operational differences in other settings. Broader implementation would require multicenter validation.

Reply: We fully agree with the Reviewer and mentioned this issue at the end of the Discussion, as follows:

A multicenter implementation of the proposed diagnostic algorithm in national and international NBS programs is recommended, to confirm its feasibility and accuracy across different populations and operational settings”. (page 13, lines 513-515).

-While the clinical necessity of early detection is emphasized, the paper does not address the cost-effectiveness of adding MLD to NBS panels, which is a key consideration for policy decisions. The authors should Incorporate a discussion or analysis of the economic implications of MLD NBS, including laboratory costs, follow-up, and potential savings from early intervention.

Reply: We agree with the Reviewer that the economic implications are of great relevance for healthcare systems and policy makers. While a full cost-effectiveness analysis falls outside the aims of this study, we now mentioned this key aspect in the Discussion, as follows:

The cost of the first-tier screening test using a mass spectrometry-based assay is relatively low when integrated into an existing NBS platform; based on our project estimates, these additional costs amount to approximately €5 per test, considering reagents, instrument servicing, consumables, and laboratory personnel. Moreover, while follow-up procedures such as enzyme assays, genetic testing, and neuroimaging may increase healthcare costs, the diagnostic algorithm based on a multi-tier approach has been shown to result in a low false positive rate, thereby reducing the burden of unnecessary follow-up. On the other hand, MLD is a progressive and devastating neurodegenerative disease with a high cost burden when diagnosed after symptom onset. Early treatment can reduce the need for intensive medical care, long-term hospitalization, mobility aids, and specialized education, leading to significant long-term savings for healthcare systems and families, as demonstrated in studies on the economic implications of MLD NBS, [ref 19. Bean, K.; Int. J. Neonatal Screen. 2024]” (pages 12-13, lines 443-457).

- The authors should discuss the impact of diagnosing the late-onset form of MLD via newborn screening, which would necessitate lifelong monitoring, particularly given the limited treatment options currently available.

Reply: It is true that, to date, late juvenile MLD cases are not eligible to gene therapy, and thus might only marginally benefit from pre-symptomatic diagnosis at NBS. However, we believe that in the next future also these patients might be treated with gene therapy, and a phase III clinical trial is ongoing on this disease phenotype [NCT04283227]. Pending the results of this trial, early detection of late juvenile MLD by NBS has the potential to revolution the disease course of these patients.

We now addressed this aspect also in the manuscript Discussion.